# Pro-inflammation and clinical correlates of unsuppressed HIV-viral load in children living with perinatally-acquired HIV 1 in Zambia

John Nzobokela[1,2], Lweendo Muchaili[1,2], Kingsley Kamvuma[1], Bislom C. Mweene[1,2], Situmbeko Liweleya[1,2], Sydney Mulamfu[1,2], Benson M. Hamooya[1,2], Sepiso K. Masenga[1,2]*

**1** Mulungushi University, School of Medicine and Health Sciences, Livingstone, Zambia, **2** Livingstone Center for Prevention and Translational Science, Livingstone, Zambia

* sepisomasenga@gmail.com, sepisomasenga@lcpts.org

## Abstract

### Background

Children living with perinatally-acquired HIV-1 (CPHIV) face significant health challenges despite advancements in antiretroviral therapy (ART). This study aimed to determine the association between unsuppressed HIV RNA viral load (VL), proinflammatory markers, sociodemographic, and clinical factors among CPHIV attending routine ART clinic at Arthur Davison Children's Hospital (ADH), Ndola, Zambia.

### Methods

We conducted a cross-sectional study on 135 CPHIV, aged 2−18 years, who had been on ART for over 12 months. Sociodemographic, clinical, and laboratory data were collected using a standardized questionnaire and a data collection form. The primary outcome was unsuppressed HIV RNA VL defined as a viral load greater than 1000 copies/mL of HIV-1 RNA after at least 6 months of ART treatment. Bivariate and multivariate logistic analyses were conducted to assess associations with unsuppressed HIV viral load.

### Results

Overall median (Q1 - Q3) age was 15 years (12–17) and 59.3% were male. The proportion of CPHIV with unsuppressed HIV VL was 15.6% (n = 21, 95% confidence interval (CI): 9.9–22.8%). Factors associated with unsuppressed VL in multivariate logistic regression were poor adherence to ART (missing two or more doses in 2 weeks) (adjusted OR (AOR) = 14.96; 95% CI: 2.39–93.49, p = 0.004) and lower CD4 count (AOR; 0.99, 0.99–1.00, p = 0.026). Proinflammatory markers tumor necrosis factor-alpha (TNF-α) (p = 0.196) and D-dimer (p = 0.709) did not differ between the suppressed and unsuppressed CPHIV.

**Data availability statement:** All relevant data are within the manuscript and its Supporting Information file S2.

**Funding:** The author(s) received no specific funding for this work.

**Competing interests:** The authors have declared that no competing interests exist.

## Conclusion

One in six children with perinatally acquired HIV in Ndola, Zambia, had unsuppressed viral load, which was associated with poor ART adherence and lower CD4 counts. Proinflammatory markers, TNF-α and D-dimer, showed no significant differences between suppressed and unsuppressed groups, suggesting they may not be reliable indicators of viral control. Enhanced adherence support and further research on immune dysregulation are needed.

## Introduction

Perinatally-acquired HIV (PHIV) is the transmission of HIV from an HIV-positive mother to the child during pregnancy, childbirth, or breastfeeding [1]. Children (0–19 years) with unsuppressed viral load (VL) (VL > 1000 copies/ml of HIV1 RNA) have a higher risk of morbidity and mortality [2,3]. Studies associate this phenomenon with persistent inflammation, leading to kidney and cardiovascular diseases, as well as other age-related chronic diseases [4–6].

In Zambia, about 58,000 children live with HIV and 71% of them are currently on ART [7]. According to a survey in the central province of Zambia, unsuppressed VL among children aged 15 years and below is 19% [8]. Children living with PHIV (CPHIV) often have a higher prevalence of unsuppressed VL than adults [9,10]. A recent review in Sub-Saharan Africa (SSA) found that 29% of them experienced virological failure, with predictors including drug resistance, poor adherence to antiretroviral therapy (ART), high VL at ART initiation, lack of primary caregivers, and concurrent tuberculosis treatment [10].

Persistent inflammation and various sociodemographic and clinical factors contribute to increased viremia. Commonly studied pro-inflammatory markers include interleukin-6 (IL-6), tumor necrosis factor-α (TNF-α), C-reactive protein (CRP), and D-dimer [6,11]. However, studies present conflicting findings on whether achieving VL suppression correlates with reduced proinflammatory responses in CPHIV. Some evidence suggests persistent inflammation despite viral suppression [12–14], while Dirajlal-Fargo et al. and Fitzgerald et al. propose an improvement in proinflammatory markers with suppression as children age [5,15].

Few studies have explored pro-inflammatory markers and sociodemographic and clinical factors associated with unsuppressed VL in CPHIV. Moreover, routine evaluation of these markers is lacking in African clinical settings, highlighting the importance of this study. This study aimed to identify proinflammatory markers, sociodemographic and clinical factors associated with unsuppressed VL among CPHIV attending the routine HIV clinic at Arthur Children's Hospital.

## Methods

### Study design and setting

We conducted a cross-sectional study at Arthur Davison Children's Hospital (ADH) involving CPHIV attending routine ART between 12th March 2024 and 30th June 2024.

ADH is the largest referral children's hospital in the Copperbelt province of Zambia that offers ART and general medical services to the community, with approximately 356 CPHIV enrolled in HIV treatment and management services.

## Study participants and eligibility

Following assent, we recruited CPHIV who had been on integrase strand transfer inhibitors (INSTI)-based ART regimen for at least one year. Perinatal HIV infection was determined from medical records using SmartCare and/or maternal history. Older children initiating ART after age 5 were included only if this was clearly documented in their medical records and no alternative route of HIV transmission was reported. Participants with evidence of self-reported or documented opportunistic or acute infections (malaria, syphilis, hepatitis C and B, tuberculosis, helminthiasis, pneumonia, meningitis) and diarrhea in the last month, as well as adolescents who were pregnant, were excluded from the study.

## Sample size determination

We used OpenEpi online software (sample size for a proportion or descriptive study) to compute a total sample size of 135. Based on the 2022 UNAIDS Supporting an AIDS-Free Era survey for the Central Province of Zambia,19% unsuppressed VL was taken to calculate the sample size to represent our study population of 356 [8].

## Study variables

The primary response variable was unsuppressed VL, defined as having a VL greater than 1000 copies/mL of HIV-1 RNA after at least 6 months of ART treatment [16]. Independent variables included sociodemographic characteristics such as age, sex, level of education attained, age at ART initiation, primary caregiver, HIV disclosure status, and support group. In this study, HIV disclosure was defined as informing children and adolescents about their HIV diagnosis. For children aged ≥5 years, this included providing comprehensive knowledge of their condition and treatment or sharing limited information without explicitly using the term "HIV," while ensuring they understand the need for medication. Non-disclosure was defined as keeping the HIV diagnosis entirely secret, with the child or adolescent remaining unaware of their illness and the reason for taking medication [17].Clinical factors included body mass index (weight in kg/height in meters$^2$), duration of ART, years, nutritional status, ART regimen substitution, TB prophylaxis, treatment for tuberculosis, adherence to ART (missing two or more doses in 2 weeks), reasons for missing doses, medication side effects experienced (Nausea/vomiting, Diarrhoea, Headache, Vivid dreams, Insomnia) and CD4 (cells/µL).

## Data collection

Assent was obtained from the participants, and written informed consent was obtained from the guardians, and data was collected from both the guardians and health records (SmartCare and Data Intensive Systems and Applications) using a structured questionnaire and data collection form, respectively. Whole blood samples were collected in ethylenediaminetetraacetic acid (EDTA) containers from all children for VL testing, as well as for full blood and differential counts and CD4 + count measurements. VL was measured using the Cobas 6800 system, with a detection limit of 20 copies/ml. Full blood and differential counts were analyzed using the Sysmex XN-1000 hematology analyzer (Sysmex Corporation, Kobe, Japan), and CD4 + T cells were measured using the BD FACSCOUNT system (Becton Dickinson and Company, California, USA). Plasma was separated within 12 hours and stored at −20°C until further analysis of pro-inflammatory markers in the research laboratory. Markers such as D-Dimer were analyzed using the iChroma- II (Boditech Med Incorporated, Republic of Korea), with a working range of 50–10,000 ng/mL, while TNF-α was measured using ELISA (IgG) testing (Bio Tek EL x800; Bio Teck instruments INC, Highland Park, USA). The samples were processed according to the manufacturer's instructions and laboratory standard operating procedures (SOPs) [18].

## Data analysis

Data were analyzed using SPSS version 22 for both descriptive and inferential analysis. Categorical variables were summarized as frequencies and percentages, and continuous variables as medians with interquartile ranges (IQR) due to non-normal distribution, confirmed by Q-Q plots and Shapiro-Wilk test. The Mann-Whitney test was used to compare medians of continuous variables between suppressed and unsuppressed VL groups, while the Pearson chi-square test assessed associations between categorical variables. Logistic regression was employed to identify factors associated with unsuppressed VL; variables with a p-value < 0.05 in univariate analysis were included in the multivariate analysis. Factors with $p < 0.05$ in the adjusted analysis were considered statistically significant at a 95% confidence interval.

## Ethical considerations

Ethical approval was obtained from the Mulungushi University School of Medicine and Health Sciences Research Ethics Committee (IRB: 00012281 FWA: 0002888) on 04th December 2023 and the National Health Research Ethics Board (REF: NHREB003/25/01/2024) on 25th January 2024. The authority to conduct the study was granted by the Arthur Davison Children's Hospital Administration, and all parents/guardians provided written informed consent/assent for participants. All data collected and analyzed were de-identified to ensure complete confidentiality. No data leading to the identification of the participants was abstracted during the data collection and analysis period. The study was conducted in accordance with the Declaration of Helsinki and Good Clinical Practice guidelines.

We have used the Strengthening the Reporting of Observational Studies in Epidemiology (STROBE) guidelines for reporting this observational study (Supplementary file 1).

## Results

### General characteristics of participants

Out of the 135 CPHIV attending routine ART clinic at Arthur Davison Children's Hospital (ADH), we found the proportion of unsuppressed HIV VL was 15.6% (95% CI 9.9–22.8) Fig 1. The age range was between 2–18 years old, and the median age was 15 years, with a predominance of males (59.3%). Most participants had been on ART for a median of 10 years

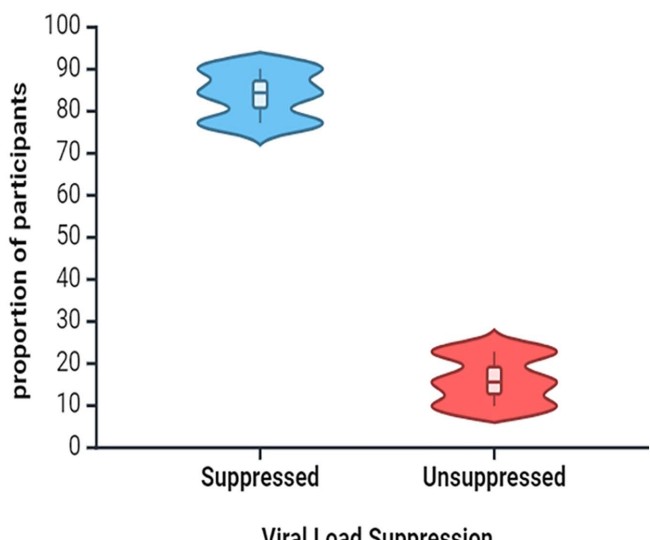

**Fig 1.** The proportion of viral load suppression.

and demonstrated good adherence, with 90.4% missing fewer than two doses. Nutritional status was predominantly normal (82.2%), and 78.5% disclosed their HIV status. The median CD4 count was 679 cells/μL, and inflammation markers showed a median TNF-α level of 42.18 pg/ml and a median D-dimer level of 432.7 μg/m Table 1.

### Sociodemographic factors associated with unsuppressed viral load

Children who were virally unsuppressed were significantly older than those who were suppressed, with a median age of 16 versus 14 years (p = 0.009). A higher proportion of children with an unsuppressed VL had attained secondary or university education (76.2%) compared to those with a suppressed VL (49.1%), showing a statistically significant association (p = 0.022). Additionally, children with an unsuppressed VL had initiated ART at a later median age of 6 years, whereas those who were suppressed started ART at a median age of 3 years (p = 0.015) Table 2

### Clinical factors associated with unsuppressed viral load

Children with unsuppressed VL had a significantly lower median CD4 count compared to those with suppressed VL, with values of 300 cells/μL versus 535 cells/μL (p < 0.001). Additionally, adherence to ART was significantly poorer in the unsuppressed group, with only 76.2% of children having good adherence compared to 93.0% in the suppressed group (p = 0.017). Although not statistically significant, there was a trend indicating higher TB prophylaxis among those with unsuppressed VL, with 90.5% receiving prophylaxis compared to 71.9% in the suppressed group (p = 0.072) Table 3.

### Hematological parameters and inflammatory surrogates associated with unsuppressed viral load

Hemoglobin levels were notably lower in the unsuppressed group (11.7 g/dL) compared to the suppressed group (12.8 g/dL, p = 0.003). Similarly, mean corpuscular hemoglobin (MCH) was significantly reduced in the unsuppressed group (25.3 pg vs. 27.5 pg, p = 0.041), and mean corpuscular hemoglobin concentration (MCHC) also showed a significant decrease (28.9 g/dL vs. 30.0 g/dL, p = 0.017). Additionally, lymphocyte counts were lower in the unsuppressed group (1.79 × 10^9/L) compared to the suppressed group (2.19 × 10^9/L, p = 0.015). The neutrophil-to-lymphocyte ratio (NLR) was higher in the unsuppressed group (1.18) than in the suppressed group (0.83, p = 0.021), and the systemic immune-inflammation index (SII) was significantly elevated in the unsuppressed group (277.87) compared to the suppressed group (209.96, p = 0.030) Table 4.

**Factors associated with unsuppressed HIV RNA viral load in logistic regression.** We evaluated factors associated with unsuppressed VL using univariate and multivariable analyses. In the unadjusted logistic regression analysis, age, level of education, adherence to ART, age at ART initiation, hemoglobin, MCH, MCHC, lymphocyte count, NLR, SII, and CD4 were significant. multivariate analysis adherence to ART and CD4 remained significantly associated with unsuppressed VL. Specifically, poor adherence to ART was associated with increased odds of having unsuppressed VL, with an AOR of 14.96 (95% CI: 2.39–93.49; p = 0.004). This indicates that individuals with poor adherence are significantly more likely to have an unsuppressed VL. Conversely, higher CD4 counts were associated with reduced odds of unsuppressed VL, with an AOR of 0.99 (95% CI: 0.99–1.00; p = 0.026), suggesting that each unit increase in CD4 count decreases the likelihood of having an unsuppressed VL by 1% Table 5.

## Discussion

The proportion of unsuppressed VL in our study was 15.6%, which is more than three times higher than the WHO's target of 5% by 2025 [19]. This indicates a significant gap in achieving viral suppression goals among CPHIV. Limited studies specifically focus on this population, as most encompass children who acquired HIV through non-perinatal means. However, our findings align with the observed proportion in the SSA region, ranging from 9.7% to 38.4% [2,20–23]. For instance, studies in Tanzania and Kenya reported slightly lower prevalences of 9.7% and 12.5%, respectively [20,22]. These differences may be attributed to their larger sample sizes, which included various modes of HIV acquisition,

**Table 1. Sociodemographic and clinical characteristics.**

| Variables | Frequency (median) | Percentage (IQR) |
|---|---|---|
| Age, *years* | 15 | 12-17 |
| Body Mass Index, *kg/m²* | 17.7 | 16.2-19.8 |
| Duration of ART, *years* | 10 | 7-13 |
| Sex | | |
| *Male* | 80 | 59.3 |
| *Female* | 55 | 40.7 |
| Level of Education | | |
| *Pre and Primary* | 63 | 61.5 |
| *Secondary and University* | 52 | 38.5 |
| Primary caregiver | | |
| *Biological* | 83 | 61.5 |
| *Non-biological* | 52 | 38.5 |
| HIV disclosure status | | |
| *disclosure* | 106 | 78.5 |
| *Non-disclosure* | 29 | 21.5 |
| Support group | | |
| *Yes* | 34 | 25.2 |
| *No* | 101 | 74.8 |
| Nutritional status | | |
| *Normal* | 111 | 82.2 |
| *Underweight* | 19 | 14.1 |
| *Overweight/Obese* | 5 | 3.7 |
| ART regimen substitution | | |
| *Yes* | 76 | 56.3 |
| *No* | 59 | 43.7 |
| TB prophylaxis | | |
| *Yes* | 101 | 74.8 |
| *No* | 34 | 25.2 |
| Treatment for tuberculosis | | |
| *Yes* | 8 | 5.9 |
| *No* | 127 | 94.1 |
| Adherence to ART | | |
| *Good (missed<2 doses in two weeks)* | 122 | 90.4 |
| *Poor (missed≥2 doses in two weeks)* | 13 | 9.6 |
| RBC Count, *10^9/L* | 4.63 | 4.26-5.11 |
| Haemoglobin, *g/dL* | 12.7 | 11.6-13.7 |
| MCV, *fl* | 86.3 | 79.0-94.2 |
| MCH, *pg* | 27.2 | 24.6-29.1 |
| MCHC, *g/dL* | 29.8 | 28.1-32.2 |
| RDW, *%* | 15.9 | 14.2-17.4 |
| WBC Count, *10^9/L* | 4.78 | 3.86-6.00 |
| Neutrophil count, *10^9/L* | 1.87 | 1.28-2.6 |
| Lymphocyte count, *10^9/L* | 2.06 | 1.6-2.63 |
| Monocyte count, *10^9/L* | 0.45 | 0.30-0.60 |
| Eosinophil count, *10^9/L* | 0.11 | 0.06-0.23 |
| Basophil count, *10^9/L* | 0.03 | 0.02-0.04 |

*(Continued)*

**Table 1.** (Continued)

| Variables | Frequency (median) | Percentage (IQR) |
|---|---|---|
| Platelets count, *10^9/L* | 267 | 277-322 |
| LMR | 4.74 | 3.55-6.68 |
| NLR | 0.85 | 0.60-1.32 |
| MLR | 0.21 | 0.14-0.28 |
| SII | 277.87 | 144.00-387.77 |
| TNF-a, *pg/ml* | 42.18 | 11.8-89.93 |
| D-dimer, *µg/mL* | 432.7 | 269.24-654.31 |
| CD4, *cells/µL* | 679 | 499-899 |

RBC, red blood cell; MCV, mean corpuscular volume; MCH, mean corpuscular hemoglobin; MCHC, mean corpuscular hemoglobin concentration; RDW, red cell distribution width; WBC, white blood cell; LMR, Lymphocyte-to-Monocyte Ratio; NLR, Neutrophil-to-Lymphocyte Ratio; MLR, Monocyte-to-Lymphocyte Ratio; SII, Systemic Immune-Inflammation.

**Table 2. Sociodemographic factors associated with unsuppressed viral load.**

| Variables | Suppressed n (%) | Unsuppressed n (%) | P value |
|---|---|---|---|
| Age, *years (IQR)* | 14 (12-16) | 16 (15-18) | **0.009** |
| Sex, n (%) | | | |
| *Male* | 69 (60.5) | 11 (52.4) | 0.485 |
| *Female* | 45 (39.5) | 10 (47.6) | |
| Level of Education | | | |
| *Pre and Primary* | 58 (50.9) | 5 (23.8) | **0.022** |
| *Secondary and University* | 56 (49.1) | 16 (76.2) | |
| Age at ART initiation, *years (IQR)* | 3 (1-5) | 6 (2-9) | **0.015** |
| Primary caregiver | | | |
| *Biological* | 70 (61.4) | 13 (61.9) | 0.965 |
| *Non-biological* | 44 (36.6) | 8 (38.1) | |
| HIV disclosure status | | | |
| *Disclosure* | 87 (76.3) | 19 (90.5) | 0.147 |
| *Non-disclosure* | 27 (23.7) | 2 (9.5) | |
| Support group | | | |
| *Yes* | 29 (25.4) | 5 (23.8) | 0.874 |
| *No* | 85 (74.6) | 16 (76.2) | |

IQR, Interquartile range; n, number of participants; %, percentage; ART, antiretroviral therapy

whereas our study focused solely on children with perinatal HIV. Conversely, higher prevalences have been reported in Uganda (23%), Côte d'Ivoire (36%) and Southern Ghana (38.4%) [2,21,23]. This may reflect the impact of the recent shift from suboptimal non-nucleoside reverse transcriptase inhibitors (NNRTIs) to optimized INSTI-based ART regimens, as outlined in the 2022 Ministry of Health, Zambia HIV guideline [24]. This underscores the need for ongoing efforts to enhance viral suppression among children on ART.

Among the clinical data collected, poor adherence to ART and low CD4 count were significantly associated with unsuppressed VL. This study revealed that children with perinatal HIV missing two or more doses of ART per two weeks had significantly higher odds of having unsuppressed VL. This aligns with findings from studies in Tanzania, where poor

**Table 3. Clinical factors associated with unsuppressed viral load.**

| Variables | Suppressed n (%) | Unsuppressed n (%) | P value |
|---|---|---|---|
| Body mass index, *kg/m²* | 17.70 (15.98-19.63) | 18.40 (16.90-22.15) | 0.110 |
| Duration of ART, *years* | 10 (7-13) | 10 (6-14) | 0.939 |
| Nutritional status (n, %) | | | |
| *Normal* | 95 (83.3) | 16 (76.2) | 0.622 |
| *Underweight* | 15 (13.2) | 4 (19.0) | |
| *Overweight/Obese* | 4 (3.5) | 1 (4.8) | |
| ART regimen substitution | | | |
| *Yes* | 6 (57.0) | 11 (52.4) | 0.694 |
| *No* | 49 (43.0) | 10 (47.6) | |
| TB prophylaxis | | | |
| *Yes* | 82 (71.9) | 19 (90.5) | 0.072 |
| *No* | 32 (28.1) | 2 (9.5) | |
| Treatment for tuberculosis | | | |
| *Yes* | 7 (6.1) | 1 (4.8) | 0.806 |
| *No* | 107 (93.9) | 20 (95.2) | |
| Adherence to ART | | | |
| *Good (< 2 doses)* | 106 (93.0) | 16 (76.2) | **0.017** |
| *Poor (≥ 2 doses)* | 8 (7.0) | 5 (23.8) | |
| Reasons for missing doses | | | |
| *Forgot* | 13 (65.0) | 2 (33.3) | 0.341 |
| *Away from home/travel* | 6 (30.0) | 3 (50.0) | |
| *Drug stockout* | 1 (5.0) | 1 (16.7) | |
| Medication side effects | | | |
| *Nausea/vomiting* | 3 (30.0) | 3 (100) | 0.103 |
| *Diarrhoea, Headache vivid dreams* | 1 (10.0) | 0 (0) | |
| *Insomnia* | 6 (60.0) | 0 (0) | |
| CD4, *cells/µL* | 535 (426-755) | 300 (164-465) | **<0.001** |

ART, antiretroviral therapy; NNRTI, Non-nucleoside reverse transcriptase inhibitor; NRTI, nucleotide reverse transcriptase inhibitor; TB, Tuberculosis.

adherence was strongly linked to unsuppressed VL [22,25]. Adherence to ART is crucial for maintaining viral suppression in individuals with HIV [26]. ART works by inhibiting the replication of the HIV [27]. Poor adherence results in suboptimal drug levels in the body, allowing the virus to continue replicating [28], which leads to higher VL [29]. Although our study found that participation in a support group was not statistically associated with VL suppression, we strongly recommend implementing targeted adherence strategies and support systems to address barriers to medication adherence and enhance overall treatment outcomes. These can include routine adherence assessment and effective usage counseling tools designed to increase ART adherence among children living with HIV.

We also found that CD4 cell count was inversely associated with the likelihood of having an unsuppressed HIV RNA VL indicated as a unit increase in CD4 cell count, the odds of having an unsuppressed VL decreased by about 1%. While this effect may seem small, it is statistically significant $p < 0.05$. This finding is consistent with studies in adults living with HIV where unsuppressed HIV VL was independently associated with lower CD4 cell count [30]. Our findings suggest that without consistent ART, the immune system remains compromised as the virus continues to attack and destroy CD4 cells [31].

**Table 4. Hematological parameters and inflammatory surrogates associated with unsuppressed viral load.**

| Variables | Suppressed n (%) | Unsuppressed n (%) | P value |
|---|---|---|---|
| RBC Count, *10^9/L* | 4.68 (4.27-5.08) | 4.33 (4.07-5.16) | 0.131 |
| Haemoglobin, *g/dL* | 12.8 (11.7-13.7) | 11.7 (9.7-12.3) | **0.003** |
| MCV, *fl* | 86.4 (79.0-95.2) | 84.2 (77.4-92.6) | 0.389 |
| MCH, *pg* | 27.5 (24.9-29.5) | 25.3 (22.8-28.1) | **0.041** |
| MCHC, *g/dL* | 30.0 (28.3-35.0) | 28.9 (26.7-30.3) | **0.017** |
| RDW, *%* | 15.7 (14.0-17.1) | 16.7 (14.9-18.9) | 0.110 |
| WBC Count, *10^9/L* | 4.88 (3.88-6.12) | 4.59 (3.58-5.82) | 0.455 |
| Neutrophil count, *10^9/L* | 1.81 (1.23-2.58) | 2.16 (1.50-3.50) | 0.228 |
| Lymphocyte count, *10^9/L* | 2.19 (1.64-2.71) | 1.79 (1.32-2.10) | **0.015** |
| Monocyte count, *10^9/L* | 0.46 (0.30-0.63) | 0.36 (0.30-0.56) | 0.199 |
| Eosinophil count, *10^9/L* | 0.12 (0.06-0.23) | 0.07 (0.03-0.19) | 0.105 |
| Basophil count, *10^9/L* | 0.03 (0.02-0.04) | 0.02 (0.01-0.04) | **0.017** |
| Platelets count, *10^9/L* | 262 (223-319) | 285 (239-346) | 0.252 |
| LMR | 4.76 (3.45-6.98) | 4.49 (3.67-5.54) | 0.397 |
| NLR | 0.83 (0.57-1.18) | 1.18 (0.77-1.72) | **0.021** |
| MLR | 0.21 (0.14-0.29) | 0.22 (0.18-0.27) | 0.397 |
| SII | 209.96 (139.36-380.21) | 277.87 (209.52-475.61) | **0.030** |
| TNF-a, *pg/ml* | 83.61 (25.56-83.61) | 71.02 (44.91-329) | 0.554 |
| D-dimer, *µg/mL* | 260 (180-438) | 400 (328-678) | 0.709 |

RBC, red blood cell; MCV, mean corpuscular volume; MCH, mean corpuscular hemoglobin; MCHC, mean corpuscular hemoglobin concentration; RDW, red cell distribution width; WBC, white blood cell; LMR, Lymphocyte-to-Monocyte Ratio; NLR, Neutrophil-to-Lymphocyte Ratio; MLR, Monocyte-to-Lymphocyte Ratio; SII, Systemic Immune-Inflammation

This weakens the body's ability to fight infections and maintain overall health. Therefore, CD4 monitoring remains valuable for managing advanced HIV disease, assessing immune recovery, and guiding clinical care [32]..

This study also revealed a relationship between older age at ART initiation and unsuppressed VL. It showed that unsuppressed VL increases as age at ART initiation increases. Although this finding is not significant in the final model after accounting for confounding variables it could be clinically important as early therapy influences both immunological and virological parameters [31]. Initiating ART at a younger age enhances the probability that CPHIV will attain normal CD4 cell levels [33]. Our finding is similar to a study in Tanzania demonstrating that each additional year in the age at which ART is initiated, the likelihood of having an unsuppressed HIV RNA VL increases by 17% [25]. Early ART initiation targets short-lived infected cells, reducing the size of viral reservoirs [27,31]. Hence it can significantly decrease the overall VL in the body. In addition, it also helps preserve the immune system by maintaining higher CD4 counts, leading to better immune function and reducing the risk of opportunistic infections [31]. Early initiation of ART is a well-recognized strategy in the management of *CPHIV*, as it improves long-term health outcomes by enhancing immune recovery, preserving CD4 counts, and reducing the risk of opportunistic infections [34,35].

It is noteworthy that a higher proportion of children with unsuppressed viral load had attained secondary or university education. This finding is likely influenced by the age distribution in our study population, as older children and adolescents are more likely to have progressed to higher levels of education. Therefore, the observed association between higher educational attainment and unsuppressed viral load may reflect the underlying effect of age rather than education itself, emphasizing the importance of considering age as a potential confounding factor in interpreting these results

**Table 5. Factors associated with unsuppressed HIV RNA viral load in logistic regression.**

| Variables | OR (95CI) | P value | AOR (95CI) | P value |
|---|---|---|---|---|
| **Sociodemographic factors** | | | | |
| Age, *years* | 1.23 (1.02-1.48) | **0.028** | 0.89 (0.66-1.20) | 0.457 |
| Sex | | | | |
| *Male* | 1 | | 1 | |
| *Female* | 1.39 (0.54-3.55) | 0.486 | 1.68 (0.47-6.05) | 0.422 |
| Level of Education | | | | |
| *Pre and Primary* | 1 | | 1 | |
| *Secondary and University* | 3.31 (1.13-9.65) | **0.028** | 3.04 (0.46-19.87) | 0.245 |
| Age at ART initiation | 1.17 (1.03-1.33) | **0.014** | 1.14 (0.96-1.35) | 0.124 |
| **Clinical factors** | | | | |
| Adherence to ART | | | | |
| *Good (< 2 doses)* | 1 | | 1 | |
| *Poor (≥ 2 doses)* | 4.41 (1.20- 14.23) | **0.024** | 14.96 (2.39-93.49) | **0.004** |
| CD4 (cells/µL) | 0.997 (0.995-0.999) | **0.002** | 0.99 (0.99-1.00) | **0.026** |
| **Hematological parameters and inflammatory surrogates** | | | | |
| Haemoglobin | 0.73 (0.60-0.90) | **0.004** | 0.84 (0.57-1.24) | 0.395 |
| MCH | 0.87 (0.78-0.97) | **0.02** | 0.99 (0.80-1.22) | 0.950 |
| MCHC | 0.79 (0.67-0.94) | **0.008** | 0.90 (0.72-1.12) | 0.367 |
| Lymphocyte count | 0.44 (0.21-0.93) | **0.032** | 0.81 (0.30-2.19) | 0.687 |
| Basophil count | 0.00(0.00-45.77) | 0.089 | | |
| NLR | 2.02 (1.05-3.89) | **0.034** | 0.68 (0.10-4.71) | 0.702 |
| SII | 1.002 (1.000-1.004) | **0.033** | 1.00(0.99-1.00) | 0.703 |

ART, ART, antiretroviral therapy; NNRTI, Non-nucleoside reverse transcriptase inhibitor; NRTI, nucleotide reverse transcriptase inhibitor; MCH, mean corpuscular hemoglobin; MCHC, mean corpuscular hemoglobin concentration; NLR, Neutrophil-to-Lymphocyte Ratio; SII, Systemic Immune-Inflammation Index

In this study, TNF-α and D-dimer levels were within or near normal ranges in the suppressed HIV group but showed higher median values in the unsuppressed group. However, neither of these differences reached statistical significance. These findings suggest that traditional markers of inflammation in this population may not reliably distinguish between those with controlled versus uncontrolled HIV viral replication. Further research is needed to explore other pathways that could contribute to immune dysregulation and inflammation in CPHIV.

The limitations of this study include its reliance on self-reported data for adherence to ART, which may introduce recall bias and social desirability bias, as consistent pharmacy refill records were not available to objectively verify adherence. This reliance limits the accuracy of adherence measures and may impact the reliability of associations observed. Additionally, the relatively small sample size may have limited the study's statistical power, reducing our ability to detect subtle differences in pro-inflammatory markers between groups. A larger sample size could provide greater precision in identifying these associations, thereby offering a more comprehensive understanding of the role of inflammatory markers in viral suppression. The study's cross-sectional design also limits its capacity to establish causality, as only associations can be inferred. However, this study also has several notable strengths. Primary data collection was used, allowing for detailed assessment of hematological and inflammatory parameters, which provided insight into the immunological factors associated with unsuppressed VL.

## Conclusion

Approximately 15.6% of children with perinatally acquired HIV in Ndola, Zambia, had unsuppressed VL, which were associated with poor ART adherence and lower CD4 counts. This highlights the need for enhanced adherence support through

personalized education, regular follow-up, and CD4 monitoring. However, proinflammatory markers, TNF-α and D-dimer, did not significantly differ between those with suppressed and unsuppressed VL, suggesting these markers may not be reliable indicators of viral control in this population. Further research is needed to investigate mechanisms contributing to immune dysregulation in children living with HIV.

## Supporting information

**S1 File. STROBE checklist.**
(DOCX)

**S2 File. Minimal dataset.**
(XLSX)

## Author contributions

**Conceptualization:** John Nzobokela, Sepiso K. Masenga.

**Data curation:** John Nzobokela, Sepiso K. Masenga.

**Formal analysis:** John Nzobokela, Sepiso K. Masenga.

**Funding acquisition:** Sepiso K. Masenga.

**Investigation:** Sepiso K. Masenga.

**Methodology:** John Nzobokela, Sepiso K. Masenga.

**Resources:** Sepiso K. Masenga.

**Software:** Sepiso K. Masenga.

**Supervision:** Lweendo Muchaili, Kingsley Kamvuma, Bislom C. Mweene, Situmbeko Liweleya, Sydney Mulamfu, Benson M. Hamooya, Sepiso K. Masenga.

**Validation:** John Nzobokela, Lweendo Muchaili, Kingsley Kamvuma, Bislom C. Mweene, Situmbeko Liweleya, Sydney Mulamfu, Benson M. Hamooya, Sepiso K. Masenga.

**Visualization:** John Nzobokela, Lweendo Muchaili, Kingsley Kamvuma, Bislom C. Mweene, Situmbeko Liweleya, Sydney Mulamfu, Benson M. Hamooya, Sepiso K. Masenga.

**Writing – original draft:** John Nzobokela, Lweendo Muchaili, Kingsley Kamvuma, Bislom C. Mweene, Situmbeko Liweleya, Sydney Mulamfu, Benson M. Hamooya, Sepiso K. Masenga.

**Writing – review & editing:** John Nzobokela, Lweendo Muchaili, Kingsley Kamvuma, Bislom C. Mweene, Situmbeko Liweleya, Sydney Mulamfu, Benson M. Hamooya, Sepiso K. Masenga.

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
