## [Decision Letter · Decision Letter 0]

13 Aug 2025

PONE-D-24-50031Pro-inflammation and clinical correlates of unsuppressed HIV-viral load in children living with perinatally-acquired HIV 1 in Zambia.PLOS ONE

Dear Dr.  Masenga,

Thank you for submitting your manuscript to PLOS ONE. After careful consideration, we feel that it has merit but does not fully meet PLOS ONE’s publication criteria as it currently stands. Therefore, we invite you to submit a revised version of the manuscript that addresses the points raised during the review process. Please submit your revised manuscript by Sep 27 2025 11:59PM. If you will need more time than this to complete your revisions, please reply to this message or contact the journal office at plosone@plos.org. Please include the following items when submitting your revised manuscript:

We look forward to receiving your revised manuscript.

Kind regards,

Jahun Ibrahim, MD, MSC, PhD

Academic Editor

PLOS ONE

Journal Requirements:

Additional Editor Comments:

The manuscript is about pro-inflammation and clinical correlates of unsuppressed HIV-viral load among children living with perinatally acquired HIV in Zambia. The manuscript is appropriate and findings from the study will certainly help in strengtehing strategies toward improving VL suppression among children. 

Please address the following observations in addition to comments raised by the two reviewers below and also ensure to review journal requirements below. 

Reviewer's Responses to Questions

**Comments to the Author**

1. Is the manuscript technically sound, and do the data support the conclusions?

Reviewer #1: Partly

Reviewer #2: Yes

2. Has the statistical analysis been performed appropriately and rigorously? 

Reviewer #1: I Don't Know

Reviewer #2: Yes

3. Have the authors made all data underlying the findings in their manuscript fully available?

Reviewer #1: Yes

Reviewer #2: Yes

4. Is the manuscript presented in an intelligible fashion and written in standard English?

Reviewer #1: Yes

Reviewer #2: Yes

5. Review Comments to the Author

Reviewer #1: Abstract is well written. However, the conclusion that a low CD4 count causes high viral load is not supported by the findings. What the authors have found is an association.

Introduction is well written

Method - it is not clear how perinatal HIV was determined/defined or assumed; this is especially important for the older children starting HIV treatment late

Results - 78% disclosed their HIV status - would be helpful to indicate to whom the disclosure was done; what proportion had their HIV status disclosed to them?

- considering that the age range included those under 5 years old, the authors should have considered disaggregating these and reporting their CD4 as a percentage and not absolute value

- in Table 1, please clarify the "adherence to ART" by adding the word "missed" before the sign < and =/> and add "in two weeks" at the end

- in line 149, it says a higher proportion of children with unsuppressed viral load had attained secondary or university education - my view is that in this population where the children with unsuppressed viral load were older, it is expected that the older children/adolescents would have attained secondary or university education. Would be good to add this to the discussion.

Discussion - line 215 - please correct prenatal to perinatal

- line 235 - the recommendation that electronic medical devices should be used to monitor adherence does not consider the downside of these which include cost and unreliability; while the use of CD4 count is recommended to monitor adherence, this has been superseded by use of viral loads which are a better measure (note that the reference cited is from 2013). However, monitoring CD4 is useful for managing advanced HIV disease and other conditions.

Conclusion - please edit "prenatal" to "perinatal". As mentioned in my review of the abstract above, the conclusion that unsuppressed viral loads were due to lower CD4 count is not supported by the findings; high viral load causes a fall in CD4 count

Reviewer #2: The manuscript is well written and congratulations to the authors for this work. However, there are a few issues to attend to;

1. Abbreviations

- Viral load (VL) - is used in the 1st paragraph of the introduction but in subsequent sections viral load is written in full, especially in the discussion.

- CPHIV is abbreviated in the 2nd paragraph of the introduction, but in discussion section 1st paragraph the abbreviation is not used - written in full as "Children living with perinatally acquired HIV"

2. There is no reference for the statement

209 may reflect the impact of the recent shift from suboptimal non-nucleoside reverse transcriptase

210 inhibitors (NNRTIs) to optimized INSTI-based ART regimens, as outlined in the 2022 Ministry of

211 Health HIV guidelines.

- assumed as Ministry of Health Zambia

3. This is a well-established concept:

251 Our conception indicates that early age at ART initiation is an essential factor to consider during

252 the care of children with prenatally acquired HIV to improve their long-term health.

- Therefore, paraphrase this sentence and make the necessary reference/s.

6. PLOS authors have the option to publish the peer review history of their article (what does this mean?). If published, this will include your full peer review and any attached files.

Reviewer #1: No

Reviewer #2: No

---

## [Author Response · Author response to Decision Letter 1]

25 Aug 2025

Response to Reviewers

PONE-D-24-50031

Pro-inflammation and clinical correlates of unsuppressed HIV-viral load in children living with perinatally-acquired HIV 1 in Zambia.

Dear Editor,

We are grateful to you and the reviewers for the careful evaluation of our manuscript and the constructive comments provided. We have carefully revised the manuscript to address each of the points raised. Below, we provide a point-by-point response to all comments. Reviewer comments are presented, followed by our responses.

Reviewer 1: Comments and Author Responses

Abstract

Comment 1: Abstract is well written. However, the conclusion that a low CD4 count causes high viral load is not supported by the findings. What the authors have found is an association.

Response:

We thank the reviewer for this important clarification. We have revised the abstract conclusion to avoid implying causality. The sentence now reads: “One in six children with perinatally acquired HIV in Ndola, Zambia, had unsuppressed viral load, which was associated with poor ART adherence and lower CD4 counts.” This revision highlights the observed association without overstating causality.

Introduction

Comment 2: Introduction is well written.

Method

Comment 3: it is not clear how perinatal HIV was determined/defined or assumed; this is especially important for the older children starting HIV treatment late.

Response:

We have revised the Methods section (Study participants and eligibility) to clarify how perinatal HIV was determined. The text now reads:“Perinatal HIV infection was determined from medical records using SmartCare and/or maternal history. Older children initiating ART after age 5 were included only if this was clearly documented in their medical records and no alternative route of HIV transmission was reported.”

Results

Comment 4: 78% disclosed their HIV status - would be helpful to indicate to whom the disclosure was done; what proportion had their HIV status disclosed to them. considering that the age range included those under 5 years old, the authors should have considered disaggregating these and reporting their CD4 as a percentage and not absolute value.

Response:

We have clarified the definition of HIV disclosure under the subheading Study variables. The revised text now reads: “In this study, HIV disclosure was defined as informing children and adolescents about their HIV diagnosis. For children aged ≥5 years, this included providing comprehensive knowledge of their condition and treatment or sharing limited information without explicitly using the term ‘HIV,’ while ensuring they understand the need for medication. Non-disclosure was defined as keeping the HIV diagnosis entirely secret, with the child or adolescent remaining unaware of their illness and the reason for taking medication.”

Comment 5: In Table 1, please clarify the "adherence to ART" by adding the word "missed" before the sign < and =/> and add "in two weeks" at the end.

Response:

We thank the reviewer for this helpful suggestion. We have revised Table 1 to clarify adherence categories. The text now reads: “Adherence to ART: Good (missed < 2 doses in two weeks); Poor (missed ≥ 2 doses in two weeks).”

Comment 6: In line 149, it says a higher proportion of children with unsuppressed viral load had attained secondary or university education. My view is that in this population where the children with unsuppressed viral load were older, it is expected that the older children/adolescents would have attained secondary or university education. Would be good to add this to the discussion.

Response:

We thank the reviewer for this insightful observation. We have now added this clarification in the Discussion section. The revised text reads: “It is noteworthy that a higher proportion of children with unsuppressed viral load had attained secondary or university education. This finding is likely influenced by the age distribution in our study population, as older children and adolescents are more likely to have progressed to higher levels of education. Therefore, the observed association between higher educational attainment and unsuppressed viral load may reflect the underlying effect of age rather than education itself, emphasizing the importance of considering age as a potential confounding factor in interpreting these results.”

Discussion

Comment 7: line 215 – please correct “prenatal” to “perinatal.”

Response:

We appreciate the reviewer for pointing out this error. We have corrected the term accordingly. The sentence now reads: “This study revealed that children with perinatal HIV missing two or more doses of ART per two weeks had significantly higher odds of having unsuppressed VL.”

Comment 8: (Line 235): The recommendation that electronic medical devices should be used to monitor adherence does not consider the downside of these, which include cost and unreliability; while the use of CD4 count is recommended to monitor adherence, this has been superseded by use of viral loads which are a better measure (note that the reference cited is from 2013). However, monitoring CD4 is useful for managing advanced HIV disease and other conditions.

Response:

We thank the reviewer for this important observation. We have revised the manuscript. The previous recommendation to use CD4 as a primary adherence measure has been removed. We now state:“Therefore, CD4 monitoring remains valuable for managing advanced HIV disease, assessing immune recovery, and guiding clinical care.”

Conclusion

Comment 9: Please edit “prenatal” to “perinatal.” As mentioned in my review of the abstract above, the conclusion that unsuppressed viral loads were due to lower CD4 count is not supported by the findings; high viral load causes a fall in CD4 count.

Response:

We thank the reviewer for this observation. We have corrected “prenatal” to “perinatal” throughout the manuscript. The conclusion in the abstract and discussion has been revised to reflect an association rather than causation. The revised sentence now reads:

“Approximately 15.6% of children with perinatally acquired HIV in Ndola, Zambia, had unsuppressed VL, which were associated with poor ART adherence and lower CD4 counts.”

Reviewer 2: Comments and Author Responses

Comment 1: The manuscript is well written and congratulations to the authors for this work. However, there are a few issues to attend to;

1. Abbreviations - Viral load (VL) - is used in the 1st paragraph of the introduction but in subsequent sections viral load is written in full, especially in the discussion.

- CPHIV is abbreviated in the 2nd paragraph of the introduction, but in discussion section 1st paragraph the abbreviation is not used - written in full as "Children living with perinatally acquired HIV"

Response:

We thank the reviewer for this helpful observation. We have ensured consistency by using the abbreviation Viral load (VL) throughout the manuscript. Similarly, we have replaced the full phrase “Children living with perinatally acquired HIV” with the abbreviation CPHIV in the discussion.The revised sentence now reads: “This indicates a significant gap in achieving viral suppression goals among CPHIV.

Comment 2: There is no reference for the statement

209 may reflect the impact of the recent shift from suboptimal non-nucleoside reverse transcriptase

210 inhibitors (NNRTIs) to optimized INSTI-based ART regimens, as outlined in the 2022 Ministry of

211 Health HIV guidelines. - assumed as Ministry of Health Zambia

Response

Response:

We have clarified the reference by adding “Zambia” between Ministry of Health and HIV guidelines. The text now reads: “may reflect the impact of the recent shift from suboptimal non-nucleoside reverse transcriptase inhibitors (NNRTIs) to optimized INSTI-based ART regimens, as outlined in the 2022 Ministry of Health, Zambia HIV Guidelines.” We have also added the citation: Zambia Consolidated Guidelines for Treatment and Prevention of HIV Infection 2022.

Comment 3. This is a well-established concept:

251 Our conception indicates that early age at ART initiation is an essential factor to consider during

252 the care of children with prenatally acquired HIV to improve their long-term health.

- Therefore, paraphrase this sentence and make the necessary reference/s.

Response:

We thank the reviewer for this suggestion. We have paraphrased the sentence for clarity and accuracy. The revised text now reads: “Early initiation of ART is a well-recognized strategy in the management of CPHIV, as it improves long-term health outcomes by enhancing immune recovery, preserving CD4 counts, and reducing the risk of opportunistic infections.” Relevant references supporting this statement have also been added

---

## [Decision Letter · Decision Letter 1]

24 Sep 2025

Pro-inflammation and clinical correlates of unsuppressed HIV-viral load in children living with perinatally-acquired HIV 1 in Zambia.

PONE-D-24-50031R1

Dear Dr.Masenga

We’re pleased to inform you that your manuscript has been judged scientifically suitable for publication and will be formally accepted for publication once it meets all outstanding technical requirements. Reviewer 2 provided minor edits that have to do with punctuations and abbreviations, please take note of them and submit a revised copy that addresses these edits,

Kind regards,

Ibrahim Jahun, MD, MSC, PhD

Academic Editor

PLOS ONE

Additional Editor Comments (optional):

Reviewer #1:

Reviewer #2:

Reviewers' comments:

Reviewer's Responses to Questions

**Comments to the Author**

1. If the authors have adequately addressed your comments raised in a previous round of review and you feel that this manuscript is now acceptable for publication, you may indicate that here to bypass the “Comments to the Author” section, enter your conflict of interest statement in the “Confidential to Editor” section, and submit your "Accept" recommendation.

Reviewer #1: All comments have been addressed

Reviewer #2: All comments have been addressed

2. Is the manuscript technically sound, and do the data support the conclusions?

Reviewer #1: Yes

Reviewer #2: Yes

3. Has the statistical analysis been performed appropriately and rigorously? 

Reviewer #1: Yes

Reviewer #2: Yes

4. Have the authors made all data underlying the findings in their manuscript fully available?

Reviewer #1: (No Response)

Reviewer #2: (No Response)

5. Is the manuscript presented in an intelligible fashion and written in standard English?

Reviewer #1: (No Response)

Reviewer #2: Yes

6. Review Comments to the Author

Reviewer #1: (No Response)

Reviewer #2: The authors need to attend to very minor issues raised:

I have the following minor edits;

Introduction

1. Line 63: …..HIV clinic at Arthur DAVISON Children’s Hospital (ADH). – Davison is omitted.

Methods

1. Line 67: use abbreviation – ADH as above.

Results

2. Line 144: Arthur Davison Children’s Hospital is already abbreviated as ADH.

Discussion

1. Line 235: Delete “one” full stop.

2. Line 251: ----viral load, abbreviate as VL

7. PLOS authors have the option to publish the peer review history of their article (what does this mean?). If published, this will include your full peer review and any attached files.

Reviewer #1: No

Reviewer #2: **Yes: **Evans Mwila Mpabalwani

---

## [Editor Report · Acceptance letter]

PONE-D-24-50031R1

PLOS ONE

Dear Dr. Masenga,

I'm pleased to inform you that your manuscript has been deemed suitable for publication in PLOS ONE. Congratulations! Your manuscript is now being handed over to our production team.

Kind regards,

on behalf of

Dr. Ibrahim Jahun

Academic Editor

PLOS ONE